# Understanding Developmental Cell Death Using *Drosophila* as a Model System

**DOI:** 10.3390/cells13040347

**Published:** 2024-02-16

**Authors:** Ruchi Umargamwala, Jantina Manning, Loretta Dorstyn, Donna Denton, Sharad Kumar

**Affiliations:** 1Centre for Cancer Biology, University of South Australia and SA Pathology, Adelaide, SA 5001, Australia; jantina.manning@unisa.edu.au (J.M.); loretta.dorstyn@unisa.edu.au (L.D.);; 2Faculty of Health and Medical Sciences, The University of Adelaide, Adelaide, SA 5005, Australia

**Keywords:** drosophila, cell death, apoptosis, caspases, autophagy, autophagy-dependent cell death

## Abstract

Cell death plays an essential function in organismal development, wellbeing, and ageing. Many types of cell deaths have been described in the past 30 years. Among these, apoptosis remains the most conserved type of cell death in metazoans and the most common mechanism for deleting unwanted cells. Other types of cell deaths that often play roles in specific contexts or upon pathological insults can be classed under variant forms of cell death and programmed necrosis. Studies in *Drosophila* have contributed significantly to the understanding and regulation of apoptosis pathways. In addition to this, *Drosophila* has also served as an essential model to study the genetic basis of autophagy-dependent cell death (ADCD) and other relatively rare types of context-dependent cell deaths. Here, we summarise what is known about apoptosis, ADCD, and other context-specific variant cell death pathways in *Drosophila*, with a focus on developmental cell death.

## 1. Introduction

Maintaining the correct cell numbers via cell division and eliminating harmful or redundant cells is crucial for the fitness of all metazoans [1]. Cells can die from different mechanisms such as accidental cell death (ACD) that can induce necrosis, or through genetically controlled methods of regulated cell death (RCD) [2]. More specifically, programmed cell death (PCD) is a specific subset of RCD that occurs exclusively under physiological conditions, with apoptosis being the most frequently used and conserved cell death pathway [3]. However, with the emergence of context-specific, non-apoptotic modes of cell deletion such as autophagy-dependent cell death (ADCD), *Drosophila* has prevailed as an excellent model system to better understand the genetic, molecular, and regulatory mechanisms underlying the unconventional forms of developmental cell death. Herein, we review our knowledge of the conserved apoptotic machinery in *Drosophila* and highlight tissue-specific modes of cell death, such as ADCD and other variant cell deaths, which support development and organismal homeostasis.

## 2. Evolutionarily Conserved Apoptotic Cell Death

The genetic machinery of apoptosis was first discovered in the nematode, *Caenorhabditis elegans*, where 131 of the 1090 cells generated during the development of the hermaphrodite worm died by this process, even though the extra cells had no effect on the viability of the worm [4,5]. Pioneering morphological and genetic studies in *C. elegans* initiated by Sulston, Horvitz, and Brenner lead to the discovery of the genetics of PCD execution that underpins the core apoptosis machinery in metazoans [4,6]. The primary apoptosis machinery in *C. elegans* comprises four cell death genes/proteins. Cell death abnormal (CED)-9 is crucial for cell survival, whereas the other three, EGL-1, CED-4, and CED-3, are required for executing cell death. The BH3-only protein, EGL-1, initiates cell death by sequestering CED-9, allowing the adaptor CED-4 to activate CED-3, a cysteine protease that cleaves target proteins, resulting in apoptosis and the clearance of the cell corpse [7,8,9,10]. This elegant pathway, while far more complex, is essentially conserved in mammals, where BH3-only, EGL-1-like proteins initiate apoptosis by blocking the survival function of CED-9-like BCL-2 family members to initiate APAF-1 (CED-4-like)-dependent caspase-9 (CED-3 like) activation (Figure 1) [11]. Although apoptosis signalling in mammals can also be initiated by extrinsic mechanisms mediated by the tumour necrosis factor receptor (TNFR) family, all apoptotic cell deaths in mammals involve the caspase, cysteine protease (CED-3) family of enzymes which are initially activated using an activating platform (apoptosome or DISC) that is functionally analogous to the CED-4 complex (the CED-4 apoptosome) (Figure 1) [11,12].

## 3. The Apoptotic Machinery in *Drosophila*

*Drosophila melanogaster* was first adopted as a model organism more than 110 years ago by Thomas Hunt Morgan to study the generational inheritance of genetic information [13]. Since then, *Drosophila* has played a critical role in understanding the genetics of many complex biological pathways which are evolutionarily conserved across phylogeny. The well-defined developmental stages in *Drosophila* comprising embryo, larva, pupa, and adult have allowed genetic studies to explore molecular pathways that drive spatiotemporal regulation of development, including developmental cell death.

The main apoptotic machinery in *Drosophila* includes the conserved family of caspases, although the upstream mechanisms that regulate the activation of caspases are divergent from *C. elegans* and mammals (Figure 1). The three pro-apoptotic genes, *grim*, *reaper* and *head involution defective* (*hid*) (also known as *RHG* genes that are transcribed on the H99 locus) act as initiators of apoptosis, with the loss of any individual gene resulting in suppression of cell death [14,15,16]. Antagonising RHG are the *Drosophila* inhibitor of apoptosis proteins (IAPs), Diap1 and Diap2. Diap1 acts as the major cell death inhibitor and blocks caspase activation by utilising its E3 ligase function to catalyse transfer of ubiquitin moieties to Dronc, thus resulting in its proteasomal degradation [17,18]. On the other hand, the binding of RHG proteins to Diap1 results in Diap1 autoubiquitination and degradation, which triggers rapid and spontaneous caspase activation and apoptosis [17,19].

There are seven caspases in *Drosophila* of which Dronc (*Drosophila* Nedd2-like caspase) is the sole apoptosis initiator caspase that has the most sequence homology to mammalian caspase-2 but is most functionally similar to CED-3 in *C. elegans* and caspase-9 in mammals [20,21]. The two effector caspases, Drice and Dcp-1, require activation by active Dronc and are functionally similar to mammalian caspase-3 and caspase-7 [22]. The caspase Dredd is partially similar to mammalian caspase-8 and caspase-10; however, its function is mainly attributed to mediating innate immune responses [23,24]. The functions of the other three caspases, Strica, Decay and Damm, are mostly apoptosis independent. Strica has a long amino-terminal Ser/Thr-rich region which remains functionally undefined, whereas Decay and Damm are similar to caspase-3 and caspase-7 but their deletion has no effect on cell death, suggesting a level of redundancy in the *Drosophila* cell death machinery [25,26,27,28].

Analogous to CED-4-mediated activation of CED-3 in *C. elegans* and APAF-1-dependent caspase-9 activation in mammals, Dronc activation requires the *Drosophila* APAF-1-related killer (Dark) apoptosome [29]. Although sharing functional similarities, the octameric CED-4 and Dark apoptosomes, as well as the heptameric APAF-1 apoptosome, assemble and activate caspases using different molecular mechanisms (discussed in detail in [30]). Also, unlike in mammals, where cytochrome *c* release from the mitochondria promotes the formation of the apoptosome and activates caspase-9, CED-4 and Dark apoptosomes have no such requirement [30]. In all cases, procaspase recruitment to the apoptosome occurs via homotypic interactions between conserved caspase activation and recruitment domains (CARDs) in these proteins; however, the stoichiometry is vastly different between species [31]. For example, the CED-4 apoptosome binds two molecules of CED-3 and facilitates CED-3 autocatalytic activity [32]. Similarly, the structure of the Dark apoptosome is also octameric, however eight molecules of Dronc are recruited in Dark [33]. Conversely, the heptameric assembly of APAF-1 binds three to four molecules of procaspase-9 [30,34]. Nonetheless, the principal role of this complex is to facilitate proximity-induced autoactivation of procaspase (zymogen) molecules [35].

*Drosophila* has two BCL-2-like proteins, Debcl and Buffy, both of which are more similar to the mammalian BAX-like protein, BOK, than other members of the prosurvival BCL-2 family proteins [36]. In overexpression and RNAi experiments, Debcl appears to serve a proapoptotic function, whereas Buffy suppresses cell death [37,38,39]. However, neither of these proteins appear essential for apoptosis or cell survival, as single or combined (double) mutants of *Debcl* and *Buffy* are viable and do not modulate PCD [40]. Recently, a BH3-like protein has also been described based on sequence homology to the conserved BH3 motif [41]. This protein, Sayonara, can complex with Buffy and Debcl and induce apoptosis when overexpressed; however, *synr* mutants are viable without adverse developmental or cell death phenotypes [41]. Thus, like Buffy and Debcl, the physiological function of Sayonara in the core apoptosis machinery remains ambiguous. It is possible that in *Drosophila*, BCL-2 family members have evolved to have context-dependent function(s) outside the main apoptosis pathway. However, *Drosophila*, unlike *C. elegans* and mammals, relies on RHG proteins to initiate apoptosis and Diap1 to inhibit caspase activation. The mammalian BCL-2 family members (including proapoptotic BH3-only proteins, prosurvival BCL-2 proteins, and pore forming tBID, BAX, and BAK) are mainly involved in controlling the release of cytochrome *c* from the mitochondria that is required for APAF-1 apoptosome formation and caspase-9 activation [42]. As cytochrome *c* is not required for apoptosome formation and caspase activation in *Drosophila*, it is likely that this specific apoptosis regulatory function for BCL-2 proteins evolved much later.

## 4. Autophagy Machinery in *Drosophila*

Originally established as a cell degradation and recycling process, autophagy is also associated with context-dependent cell death processes. Autophagy is an evolutionarily conserved response to adverse cellular events such as starvation, hypoxia, and microbial infection, protecting cells from stress-induced death [43]. This occurs via sequestration of redundant or detrimental substrates (including damaged organelles) into double-membraned vesicular structures known as autophagosomes, followed by their degradation via lysosomal-mediated pathways [44]. The recycling of substrates through autophagy releases energy and monomers of complex biomolecules (fatty acids and amino acids) into the intracellular environment, which can be utilised by cells to ensure cellular homeostasis when compromised [45]. While autophagy operates constitutively under basal conditions to maintain cellular health, elevated levels occur when cells face adversities [46].

The core machinery of autophagy was first discovered through genetic screens performed in *Saccharomyces cerevisiae* (yeast), unveiling 41 genes encoding important autophagy regulatory components [47,48]. These genes are highly conserved in mammals and *Drosophila* [48,49,50].

In *Drosophila*, autophagy is activated upon the formation of the Atg1 initiation complex, comprising Atg1 (a serine/threonine kinase, homologous to mammalian ULK1), Atg13, Atg17 (FIP200 in mammals) and Atg101 [49,51] (Figure 2). The Atg1 complex translocates to the isolation membrane of the developing autophagosome (also known as the phagophore), where it recruits Atg9 as well as the class III phosphatidylinositol 3-kinase (PtdIns3K) complex [52]. The PtdIns3K complex, consisting of vacuolar protein sorting (VPS), VPS34 and VPS15, Atg6 (Beclin 1 in mammals) and Atg14, phosphorylates phosphatidylinositol to phosphatidylinositol-3-phosphate (PI(3)P), which then promotes the nucleation of the autophagosomal membrane (Figure 2) [53,54].

The expansion of the autophagosome is dependent on two ubiquitin-like conjugation systems (Figure 2). Firstly, the E1- and E2-like enzymes, Atg7 and Atg10, respectively, coordinate attachment of Atg16 to Atg5-Atg12, forming the E3-like complex, Atg5-Atg12-Ag16 [55,56]. The second ubiquitin-like system involves the tethering of Atg8a (similar to mammalian LC3 and GABARAP family members) to the autophagosomal membrane [57]. Atg8a is cleaved by the cysteine protease, Atg4, and becomes lipidated through the attachment of phosphatidylethanolamine (PE) to form Atg8a-PE by Atg7 (E1-like enzyme) and Atg3 (E2-like enzyme), the latter of which is recruited by the Atg5-Atg12-Atg16 complex to the developing autophagosome [57,58].

Finally, fusion of autophagosomes to lysosomes degrades biomaterials and is coordinated by soluble N-ethylmaleimide-sensitive factor attachment protein receptors (SNAREs) in combination with tethering factors targeting the autophagosomal Atg8a proteins and lysosomal RAB7 proteins [59].

Autophagy is tightly controlled by the conserved protein kinase, target of rapamycin (TOR), which forms the catalytic subunit of two distinct multiprotein complexes, mTORC1 and mTORC2 (mechanistic target of rapamycin) [60]. These complexes are distinguishable by their additional components—raptor, lst8, lobe (mammalian PRAS40) in mTORC1, and rict-1 (mammalian Rictor), sinh-1 (mammalian mSin1) and Lst8 in mTORC2 [61]. mTORC1 is the primary sensor of nutrient and amino acid availability, supporting cellular metabolism, homeostasis, and autophagy [62]. Although the roles of mTORC2 are yet to be fully elucidated, some functions include the regulation of metabolism and cell proliferation [62,63,64]. In nutrient-rich conditions, autophagy is downregulated by mTORC1-mediated hyperphosphorylation of Atg13 which prevents its association with Atg1. This is attenuated under starvation conditions and mTORC1 inhibition by rapamycin treatment, where dephosphorylation of Atg13 promotes the formation of the Atg1 complex [65,66]. Interestingly, an increased expression of Atg13 can promote Atg1 phosphorylation by TOR and subsequently inhibit autophagy in *Drosophila*, demonstrating dual roles for Atg13 in autophagy suppression and activation [51].

## 5. Autophagy-Dependent Cell Death (ADCD)

The primary role of autophagy is to promote cell survival under basal and adverse conditions. Challenging this dogma are several context-specific physiological processes that reveal the cytotoxic nature of autophagy.

Following the stress signalling that accompanies cell death, autophagy is often observed in cells undergoing apoptosis. However, a distinction must be made between autophagy that accompanies cell death, and autophagy that promotes cell killing. For example, in Parkinson’s disease and Danon disease, the accumulation of autophagic structures was initially regarded as a feature of cell death by autophagy, when in actuality, the increase in autophagic vacuoles was as a result of defective autophagy and not increased cell death [67,68,69].

To overcome such confusion, the Nomenclature Committee on Cell Death (NCCD) proposed the following criteria to define ADCD: (1) inhibition of autophagy must prevent cell death, (2) the process must be functionally dependent on two or more autophagy-related genes/proteins participating in ADCD, (3) and ADCD must occur independently of other cell death processes, such as apoptosis and necrosis [3,70].

## 6. Apoptosis and ADCD in *Drosophila* Development

The development of *Drosophila* is marked by drastic morphological changes from an early embryo to three larval stages, followed by the pupation and formation of the adult fly. This requires tight regulation and coordination of both apoptotic and autophagic machinery, with specific tissues relying on one or both these processes to undergo remodelling.

### 6.1. Embryogenesis

*Drosophila* embryogenesis is a highly dynamic process featuring the rapid transformation of numerous tissue structures at specific developmental stages. The earliest instance of apoptotic cell death in *Drosophila* was found to occur at stage 11 of embryogenesis [71]. Apoptosis begins in the dorsal area of the head region and leads to a retraction of the germ band, allowing dying cells along the dorsal ridge and ventral midline to facilitate closure of the dorsal tissue. Widespread cell death during neurodevelopmental stages 15 and 16 causes head involution, condensation of the ventral cord and the restructuring of brain lobes [71]. Whilst caspases are the primary drivers of these early events, late embryogenesis events, including degradation of the amnioserosa (AS) or extraembryonic tissue, rely on induction of autophagy to coordinate caspase-dependent cell death [72]. However, a subsequent study by Cormier et al. [73] demonstrated that *Atg1* mutants did not significantly attenuate caspase-dependent AS extrusion, suggesting that there is no strict requirement for autophagy induction in this process.

### 6.2. Midgut

The *Drosophila* larval midgut (anterior and posterior midgut) comprises a single layer of epithelial cells [74,75]. The proventriculus, also known as the cardia, is located anteriorly to the midgut midbody and controls passaging of food into the midgut [76]. The four arm-like structures, collectively known as the gastric caeca, branch off from the proventriculus [77].

The larval midgut undergoes drastic morphological changes during metamorphosis to give rise to the adult midgut. Despite the majority of *Drosophila* tissues dying via apoptosis, the larval midgut degrades strictly via ADCD. Beginning at −4 h relative to puparium formation (RPF), a dramatic increase in autophagy coinciding with the transcriptional upregulation of autophagy genes promotes contraction of gastric caeca, removal of the proventriculus, and condensation of the larval midgut, giving rise to the adult midgut by +12 h RPF [78,79]. Midgut histolysis remains unaffected by caspase inhibition; however, ablation of the autophagy-related genes, *Atg1*, *Atg2* and *Atg18*, significantly delays gastric caeca and midgut retraction. However, as midgut degradation is not completely blocked, this suggests potential involvement of other catabolic pathways to facilitate complete midgut degradation [78]. Regardless, the midgut remains the best example of ADCD in vivo.

### 6.3. Salivary Glands

The salivary glands are large tubular structures extending from the larval mouth which undergoes abrupt degradation during metamorphosis. Following midgut histolysis, salivary gland cell death is initiated at +10 h RPF and is degraded by +15 h RPF [80,81]. Increased nuclear permeability of salivary gland cells to acridine orange observed at +14 h RPF confirmed the involvement of apoptotic machinery in salivary gland degeneration; however, only a partial block in histolysis was achieved upon expression of the apoptosis inhibitor, baculovirus p35 [81]. Subsequent studies showed that intact apoptotic and autophagic machinery was required to effectively induce salivary gland cell death, with the loss of either pathway delaying tissue removal [82]. This suggests that the histolysis of the salivary glands is dependent on both autophagy and apoptosis.

### 6.4. Fat Body

The fat body, analogous to vertebrate adipose tissue, is an essential component of *Drosophila* which serves an important role in meeting metabolic demands during larval and adult stages of life [83]. Prior to metamorphosis, wandering larvae stop feeding in order to undergo pupation. In response to starvation, robust levels of autophagy promote fat body shrinkage to provide energy for survival [84]. Additionally, *Drosophila* must rely on stored energy to fuel pupal-to-adult transition [85]. Individualisation of fat body cells during metamorphosis causes their redistribution within the pupa, and these are progressively eliminated at the early stages of life [83]. In a study by Scott et al. [86], it was shown that increasing autophagy levels by overexpressing *Atg1* resulted in caspase-dependent fat body cell death, as demonstrated by DNA fragmentation and disruption of the cytoskeleton. Thus, in the context of fat body remodelling, late larval development relies on autophagy-dependent energy production for survival, whereas during metamorphosis, autophagy precedes apoptosis as opposed to directly coordinating cellular demise. 

### 6.5. Myogenesis

The formation of *Drosophila* musculature occurs in two waves: the first wave of myogenesis during embryogenesis gives rise to mesodermal-derived somatic muscles that support larval motility, whereas the second myogenic wave initiates the formation of adult muscles responsible for flight, walking, and mating behaviours [87]. During metamorphosis, whilst most larval muscles undergo histolysis, some are retained into adulthood to facilitate proper body patterning prior to undergoing PCD [88,89]. Specifically, within the abdomen, the dorsal external oblique muscles (DEOM) are degraded during early metamorphosis at 12 h RPF, whereas the dorsal internal oblique muscles (DIOM) persist until eclosion and are removed within 24 h of adulthood [90,91]. Although increased levels of autophagy have been reported in degenerating DEOMs, knockdown of *Atg1*, *Atg5* and *Atg18* did not affect DEOM removal [92]. In contrast, overexpression of *p35* caused retention of DEOMs, signifying that while the role of autophagy in DEOM PCD is redundant, this process is largely dependent on apoptosis [92,93].

### 6.6. Neurogenesis

The formation of the *Drosophila* central nervous system (CNS) begins during embryogenesis, where neuroblasts (NBs) undergo asymmetric divisions to replenish the stem cell pool and produce ganglion mother cells (GMCs), the latter which further divide to create neurons and glial cells [94]. Whilst a majority of NBs cease to proliferate at +30 h RFP during pupal development, NBs within a region of the brain known as the mushroom body (MB) continue dividing up to 10 h prior to eclosion of the adult fly, following which they undergo cell death [95]. Elimination of MB NBs is required to complete neurogenesis, and, in a similar manner to the salivary glands, this is dependent on both autophagy and apoptosis. Downregulation of autophagy or RNAi-mediated inhibition of *RHG* genes results in persistence of MB NBs for up to 3 days and 7 days post-eclosion, respectively. Conversely, MB NBs survive well into adulthood upon ablation of both pathways [96].

### 6.7. Oogenesis

*Drosophila* oogenesis is a highly dynamic process involving cell-to-cell communication and maintenance of stem cell niches to support egg production [97]. Cytoblasts undergo multiple rounds of division to form an oogenic cyst consisting of 16 germ cells, of which only one cell proceeds to become an oocyte and the remaining cells develop into supporting nurse cells [98]. During the later stages of oogenesis, nurse cells expel cytoplasm-containing maternal mRNAs, organelles, and proteins into the maturing oocyte, leading to PCD of nurse cells while causing no detrimental effects to the attached oocyte [99,100]. Though there is an established apoptotic role in the removal of egg chambers during germarium and mid-oogenesis stages, recent findings suggest that this process is also triggered by nutrient deprivation [101,102]. Under starvation conditions, knockdown of *Atg1* and *Atg7*, as well as *Dcp-1* effector caspase mutants, fail to activate autophagy, causing a persistence of egg chambers [101]. Conversely, mutants of the Diap1/2 protein, *dBruce*, enhance egg chamber degradation, suggesting that while Dcp-1 can promote autophagy, dBruce operates antagonistically in this process [101,103].

## 7. Signalling Mechanisms Regulating ADCD in *Drosophila*

### 7.1. Ecdysone Signalling

The transition of the *Drosophila* larvae to an adult fly is defined by stages of larval molting, followed by metamorphosis during which cell death results in a structural reorganisation of obsolete larval tissues and the formation of adult tissues [104]. This is tightly regulated by the steroid hormone, 20-hydroxyecdysone (20E), in a spatiotemporal manner. Produced in the prothoracic glands from dietary cholesterol, the precursor of 20E, ecdysone, is secreted into the haemolymph and is oxidised into active 20E within the target tissues [105]. Significant pulses of ecdysone titre release occur at the late third instar, prepupal, and pupal stage, although other smaller waves of ecdysone release facilitate larval molting during the first and second instar stages [106,107,108]. Ecdysone release at the late larval stage initiates prepupa formation and degradation of the larval midgut at −4 h RPF [109]. A subsequent ecdysone titre pulse +10–12 h RPF signifies pupa development and salivary gland degradation, the latter which is complete by +15–16 h RPF. A final pulse at 30 h RPF coordinates adult development [110].

Downstream signalling through ecdysone is mediated by the heterodimeric receptor complex, ecdysone receptor/ultraspiracle (EcR/USP), which can activate a transcription of ecdysone-responsive transcription factor genes such as Broad-Complex (*BrC*), *E74*, *E75* and *E93* [111,112] (Figure 3). *E93*, in particular, was previously implicated in the degradation of salivary glands through autophagy and apoptosis [112]. However, a study by Duncan, et al. [113] showed that the allelic mutations performed by Lee and Baehrecke [111] were of the β subunit of the mitochondrial enzyme, isocitrate dehydrogenase-3 (Idh3b), not the three “type” alleles of the E93 family, *E93*^1^, *E93*^2^ and *E93*^3^. Consequently, Duncan, Kiefel and Duncan [113] demonstrate that mutations in *Idh3b* alleles prevent autophagy initiation during salivary gland degradation. Although *E93* has also been shown to activate autophagy through the downregulation of growth signalling in the removal of MB NBs [96], recent studies confirm that *E93* does not regulate expression of cell death genes in the salivary glands [114].

Larval fat bodies also rely on ecdysone receptors to regulate growth signalling and induce autophagy. Whilst the loss of ecdysone receptors prevents autophagy, inhibition of growth signalling can restore autophagy levels [115,116]. Additionally, the knockdown of *EcR* has been shown to block larval midgut degradation by reducing the transcription of autophagy-related genes, *Atg1*, *Atg13*, *Atg17* and *Atg8a*, amongst multiple others, thereby confirming its necessity in *Drosophila* ADCD [117].

### 7.2. Growth Signalling

Early *Drosophila* larval development is driven by growth signalling pathways which converge on mTORC1 (Figure 3) [118]. Insulin-like peptides (ILPs) bind to the *Drosophila* insulin receptor (dInR), causing downstream phosphorylation of the insulin receptor substrate protein, CHICO [119]. Subsequently, phosphorylation of the active subunit of *Drosophila* PI3K, Dp110, catalyses the conversion of phosphatidylinositol 4,5-biphosphate (PIP2) to phosphatidylinositol 3,4,5-triphosphate (PIP3) [120,121]. Antagonising the function of Dp110 is *Drosophila* PTEN (dPTEN) which converts PIP3 into PIP2 [122]. Acting as a secondary messenger, PIP3 recruits PDK1 and phosphorylates AKT at the plasma membrane, enabling AKT-dependent phosphorylation of *Drosophila* TSC2 (dTSC2) and subsequently maintaining the activity of mTORC1 [121,123].

As inhibition of growth signalling is required for autophagy induction, mTORC1 downregulation is essential for midgut removal by ADCD [124]. As expected, inhibition of PI3K can similarly promote midgut degradation, as does the expression of negative regulators of the PI3K pathway, d*PTEN* and *TSC1/2* [125].

Salivary glands also demonstrate the requirement for growth arrest prior to autophagy induction and histolysis [126]. The expression of *Dp110* and *AKT* results in persistent enlarged salivary glands, which undergo apoptosis- and autophagy-dependent histolysis upon removal of PIP3 and AKT from the outer membrane of salivary gland cells [82]. A similar morphological phenotype is observed when a positive regulator of PI3K signalling, Ras, is expressed in the salivary glands [82]. Another pathway that regulates growth signalling and ADCD involves the Warts (wts) family of genes, *Wts*, *Hippo* (*Hpo*), *Mob-as-tumour suppressor* (*Mats*), *Salvador* (*Sav*), *Merlin* (*Mer*) and *Expanded* (*Ex*), which are important for growth control [127]. Wts facilitates autophagy in salivary glands whereas homozygous mutation of *wts* prevents the removal of AKT from the salivary gland cell cortex during puparium formation, in turn inhibiting salivary gland degradation. This defect was shown to be rescued by ectopic expression of *Atg1*, reaffirming that growth arrest precedes autophagy-dependent salivary gland histolysis [128]. Interestingly, the knockdown of Wts family members does not impede midgut degradation, suggesting that signals regulating growth and cell death occur in tissue-specific manners [125].

### 7.3. Decapentaplegic Signalling

The bone morphogenetic protein/transforming growth factor β (BMP/TGF-β) ligand, decapentaplegic (Dpp), has been known to regulate cell proliferation, cell fate, and body patterning during embryogenesis and larval development [129,130]. Dpp exists in a heterodimeric complex consisting of the type I receptor, Thickveins (Tkv), and the type II receptor, Punt (Put). Phosphorylation of Tkv by Put triggers Tkv kinase activity, causing the phosphorylation of downstream regulators, Mothers against Dpp (Mad) and Medea (Med) [131]. Acting as a complex, Mad and Med can deregulate or activate target genes such as Brinker (Brk), a negative regulator of Dpp-dependent genes, or Schnurri (Shn), responsible for facilitating the Dpp-mediated repression of *Brk* [131,132,133].

Recent studies have uncovered that Dpp signalling is required for retention of the *Drosophila* larval midgut until it begins to undergo histolysis during the third instar stage [117]. In the presence of Dpp signalling through the expression of *Dpp* or *Tkv*, ecdysone production within prothoracic glands and transcriptional upregulation of ecdysone-responsive genes in the midgut is repressed, thereby inhibiting ADCD (Figure 3). Conversely, knockdown of *Mad* and *Med*, or blocking Dpp signalling through expression of its inhibitory SMAD protein, Dad, restored autophagy levels and accelerated midgut removal. Hence, there is convergence of multiple signalling pathways in the initiation and regulation of ADCD (summarised in Figure 3).

## 8. Alternative Cell Death Pathways in *Drosophila*

In addition to apoptosis and ADCD, non-canonical forms of cell deaths have also been reported in *Drosophila*, as summarised below.

### 8.1. Parthanatos

Parthanatos, related to the Greek term for the personification of death, “thanatos”, describes cell death involving a family of DNA repair enzymes collectively known as poly(ADP-ribose) polymerases (PARPs) [134]. PARP-1 in particular, has a variety of nuclear roles including base excision repair (BER) to single-stranded DNA break repair [135]. Whilst PARP-1 instigates a cell-protective response to acute DNA aberrations, an overaccumulation of DNA damage results in the hyperactivation of PARP-1, leading to the exhaustion of cellular energy due to a depletion of NAD+ and consequent translocation of an apoptosis-inducing factor (AIF)-DNase complex from mitochondria into the nucleus. Subsequently, AIF can trigger condensation of nuclear chromatin, disruption of mitochondrial membrane potential, externalisation of phosphatidylserine on the cell surface and cell death [136,137].

In a study by Tarayrah-Ibraheim et al. [138], it was shown that parthanatos was responsible for the elimination of primordial germ cells (PGCs) during *Drosophila* embryogenesis. Permeabilisation of lysosomal membranes releases cathepsin B, triggering mobilisation of AIF from mitochondria. Another lysosomal hydrolase, DNase II, is translocated into the nucleus by AIF and disrupts chromatin architecture, causing hyperactivation of PARP-1. A positive feedback loop between PARP-1 and AIF results in PGC death. Despite lysosomal leakage that suggests the involvement of autophagy, genetic inhibition of autophagy genes did not attenuate cell death, inferring that autophagy is not critical for this process.

### 8.2. HtrA2/Omi in Caspase-Independent Cell Death

HtrA2/Omi is a serine protease released from mitochondria upon induction of apoptosis. In a similar manner to mammalian Smac/DIABLO and *Drosophila* RHG proteins, HtrA2 can bind to and prevent IAPs from suppressing caspase-3 activity [139,140,141].

Recently, an intriguing role for HtrA2 has been unveiled in *Drosophila* germ cell death (GCD). During *Drosophila* spermatogenesis, germ cells undergo multiple rounds of synchronous divisions but remain connected by cytoplasmic bridges due to incomplete cytokinesis. Up to a quarter of differentiated spermatocytes undergo spontaneous cell death, and the rate of GCD can increase during starvation conditions to support survival of spermatogonial stem cell populations [142,143]. *Drosophila* GCD has been shown to occur independently of caspase activation [144]. Instead, the catalytic activity of HtrA2, but not its IAP-inhibiting function, controls lysosomal and mitochondrial-driven cell death by interacting with Pink1, an important modulator of mitochondrial homeostasis [144]. Furthermore, Endonuclease G (EndoG), a proapoptotic mitochondrial enzyme, is released from mitochondria to induce cell death by altering chromatin dynamics in parallel to HtrA2-mediated cell death [144,145].

## 9. Conclusions

*Drosophila* has provided the cell death community a highly accessible tool to investigate and understand tissue homeostasis and physiological processes of apoptotic and non-apoptotic forms of PCD. Combined molecular and genetic studies in *Drosophila* have provided a significant insight into the diversity of cell death pathways, including the evolutionary divergence of apoptotic machinery and the manner by which caspase activation is triggered (e.g., removal of Diap1, rather than the involvement of BH3-proteins, BCL-2 and Bax/Bak-dependent mitochondrial membrane permeabilisation). Investigations involving *Drosophila* also played a vital role in establishing autophagy as a regulator of cell survival and cell death. Whilst genetic evidence and the rationale for death by autophagy (ADCD) has been uncovered, exactly how cells die by autophagy requires further investigation. Additionally, *Drosophila* models have provided much benefit to in vivo studies into alternate forms of cell death involving non-canonical roles of apoptotic machinery, enabling clear distinctions between the context- and tissue-specific roles of cell death players in diverse physiological settings. Finally, as dysregulated apoptosis and autophagy have been implicated in various diseases (see [146,147,148]), *Drosophila* should continue to contribute knowledge relevant to both the fundamentals of cellular homeostasis in organismal development and healthy ageing.

## Figures and Tables

**Figure 1 cells-13-00347-f001:**
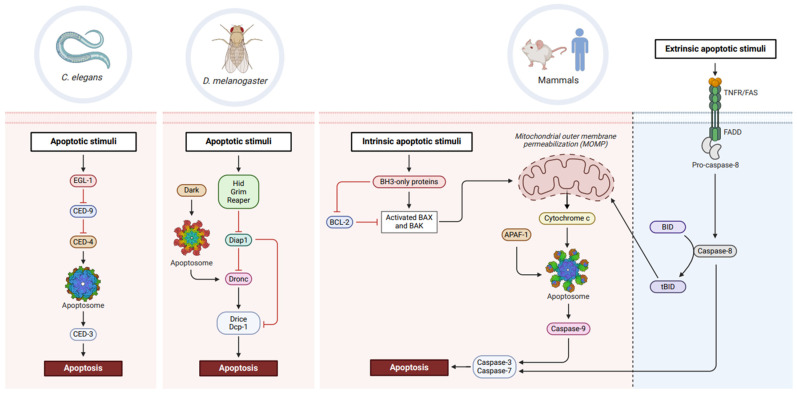
Conservation of apoptotic machinery across phylogeny. In *C. elegans*, the BH3-only protein, EGL-1, inhibits the prosurvival factor, CED-9. This releases CED-4 which forms the apoptosome and promotes autocatalytic cleavage of CED-3 into an active caspase, thereby eventuating in cell death. *Drosophila* apoptosis proceeds with RHG proteins inhibiting Diap1, leading to Dark apoptosome-mediated activation of Dronc. Activation of downstream effector caspases, Drice and Dcp-1, signals cellular demise. In mammals, BH3-only proteins inhibit BCL-2 prosurvival activity, allowing activated BAX/BAK proteins to trigger mitochondrial outer membrane permeabilisation (MOMP). Cytochrome *c* release coordinates APAF-1 apoptosome formation and enables caspase-9 activation, subsequently promoting cell death in a caspase-3 and -7-dependent manner. Extrinsic lethal stimuli recognised by death receptors (TNFR, FAS) activate caspase-8, leading to caspase-3 and -7 activation and cell death. Truncation of BID to tBID by caspase-8 can also trigger MOMP and feed into intrinsic apoptotic pathways. Created with BioRender.com.

**Figure 2 cells-13-00347-f002:**
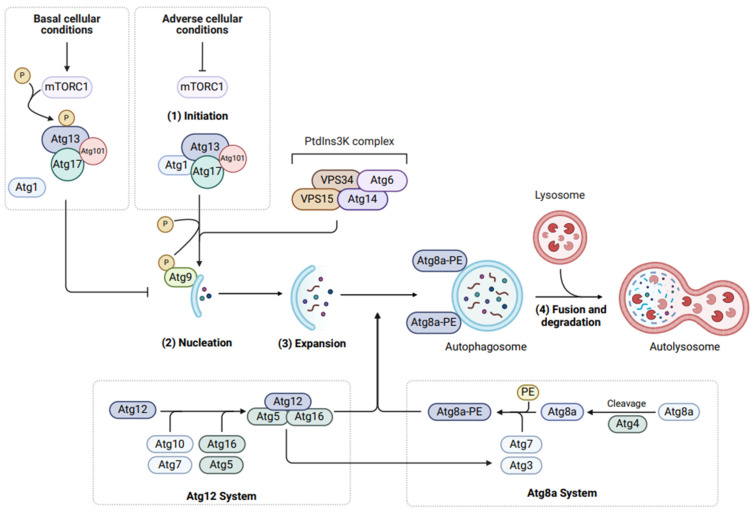
The molecular machinery of autophagy. Autophagy is an evolutionarily conserved process comprising (1) intiation, (2) nucleation, (3) expansion, and (4) fusion and degradation. Under basal or optimal cell conditions, mTORC1 remains active and phosphorylates Atg13, thereby preventing recruitment of Atg1. Cellular stressors including nutrient deprivation or energy depletion inactive mTORC1, thereby enabling formation of the Atg1 initiation complex (Atg1, Atg13, Atg17, Atg101). Nucleation occurs upon Atg1-dependent phoshophorylation and recruitment of Atg9, as well as the PtdIns3K complex (VPS15, VPS34, Atg6, Atg14). Expansion of the autophagosome relies on two ubiquitin-like conjugation systems—Atg12 and Atg8a. Conjugation of Atg16 to Atg5-Atg12 forms an E3-like complex that recruits the E2-like Atg3 to facilitate attachment of PE to Atg8a (Atg8a-PE) at the autophagosomal membrane. Fusion of autophagosomes to lysosomes relies on tethering proteins (SNAREs) and lysosomal membrane proteins (RAB7), forming an autolysosome within which lysosomal acid hydrolases degrade sequestered proteins and organelles. Created with BioRender.com.

**Figure 3 cells-13-00347-f003:**
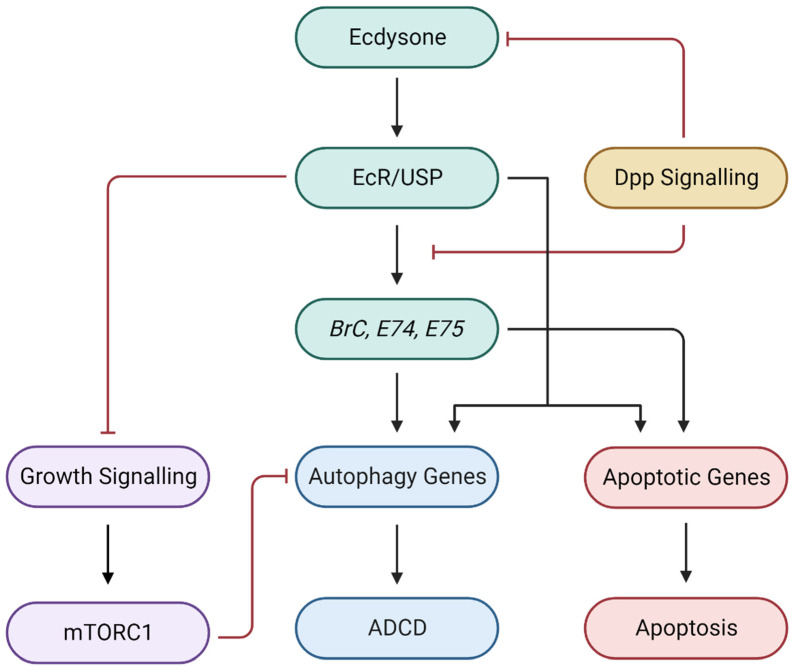
Regulatory mechanisms underlying developmental PCD in *Drosophila*. Ecdysone, produced in the prothoracic glands from dietary cholesterol, circulates in haemolymph and transcriptionally upregulates ecdysone-responsive genes (*BrC*, *E74*, *E75*) in target tissues. Subsequent transcription of autophagy and apoptotic genes coordinates tissue deletion at multiple stages of *Drosophila* development. Downregulation of PI3K signalling inhibits mTORC1, thereby activating autophagy. Dpp signalling blocks ecdysone production and transcription of ecdysone-response genes, and must therefore be downregulated to trigger ADCD. Created with BioRender.com.

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
