# Peer review of "Understanding Developmental Cell Death Using Drosophila as a Model System"

_cells, 2024, doi:10.3390/cells13040347_

Round 1

Reviewer 1 Report

Comments and Suggestions for Authors

The manuscript represents the detailed review of programmed cell death machinery in Drosophila. Authors as well compare cell death pathway in a few model organisms. The review is well written and comprehensive. In my opinion, the most valuable part is a description the specifics of cell death in particular tissues.In this regard, my only suggestion for improvement - to expand a bit a specifics of a tissue-specific pathways of cell death. For example, to provide examples of a tissues specific induction of apoptotic cell death through different RHG and ski proteins.

Reviewer 2 Report

Comments and Suggestions for Authors

The manuscript by Umargamwala et al, titled "Understanding Developmental Cell Death Using Drosophila as a Model System" presents a literature review on the various types of regulated cell deaths observable in the Drosophila model system. The authors offer concise summaries of the regulatory mechanisms of apoptosis and autophagy, accompanied by case studies featuring specific examples across different developmental stages and tissues where programmed cell death takes place. The manuscript is well-written and includes clear, informative illustrations, facilitating ease of reading and navigation. However, the Reviewer has several suggestions to further enhance the manuscript.

Major points:

1.        Since the manuscript proficiently highlights autophagy as a central theme, aligning the title more closely with its content is recommended for clarity.

2.        The case studies make this review very practical, but they wouldn’t be complete without the programmed muscle death. The massive death of larval somatic muscles during metamorphosis has been studied by many researchers, including Lawrence Schwartz, Fabio Demontis, Martin Wasser, and Norbert Perrimon among others. This topic deserves to be included in the review, especially since autophagy plays a notable role there.

3.        As a reader, the Reviewer was fascinated by the drastic morphological changes reported for midgut remodeling. However, it was not clear to what degree these changes were due to a decline in cell size rather than a decline in cell number. Interestingly, none of the cited papers (refs 78-80) had conducted cell counts within the disappearing gastric caeca, although size modulation in polyploid cells, such as enterocytes, could lead to drastic changes in gross morphology (as seen in atrophic muscles). Without a tab on cell numbers, the claims of autophagy as a direct cause of death might be somewhat overstretched. The authors might want to contemplate this issue to strengthen their conclusions.

Minor points:

1.        Reference 82 to be replaced with a better fitting Jiang C, Baehrecke EH and Thummel CT (1997) Steroid regulated programmed cell death during Drosophila metamorphosis. Development 124: 4673 – 4683.

2.        In References: some citations have active hyperlinks, while others don’t, yet others have no links at all. Please unify.

3.        Should the Bruce protein be introduced into the Drosophila apoptosis scheme (Fig 1) since it receives some attention later in the text?

4.        Line 302: A clarification is needed that Dcp-1 and Bruce have antagonistic effects on autophagy.

5.        Line 318: consider replacing “obsolete midgut” with more neutral “larval midgut”.

6.        Figure 3: should there be reciprocal arrows drawn between the boxes depicting autophagy genes and apoptotic genes?  This could be logical per ref [96] and other studies demonstrating a cross activation between autophagy and apoptosis.

7.        Line 394: Simplify the sentence on to enhance readability: "…through expressing its inhibitory SMAD protein, Dad…"

8.        Line 405: consider replacing “ranging” by “including”, which would better suit the context.

9.        In References: Not always the year of publication in the cited references is shown in bold, e.g.[4], [80]. Please unify.

10.   Please address unnecessary duplication of references that do not provide a qualitative gain, such as in the case of [78] and [79], and, similarly, [90] and [91].

11.   References: Please verify the required bibliography style of the journal for consistency in presenting journal names, either in full or as standard abbreviations.

12.   The authors are encouraged to express their opinion on whether E93 is a valid player in regulating autophagy. As of now the presented information is controversial, without a summative conclusion.

Reviewer 3 Report

Comments and Suggestions for Authors

This review primarily focuses on cell death during the developmental process in Drosophila. Firstly, an introduction is provided on the mechanisms of apoptosis in Drosophila, highlighting their similarities to apoptosis mechanisms in nematodes and mammals. Secondly, the review summarizes the process of autophagy in Drosophila, specifically elaborating on the formation of autophagosomes. The accurate determination of apoptosis and autophagic cell death is also deliberated. Subsequently, a comprehensive review is provided on cell death in specific tissues associated with Drosophila development, including embryonic formation, midgut, salivary gland, fat body, neurogenesis, and oogenesis. Furthermore, the regulation of autophagic cell death in Drosophila by ecdysone, mTORC1, and formyl peptides is also discussed. Lastly, two additional cell death pathways, including Parthanatos and HtrA2/Omi-related cell death are introduced. Overall, this review provides a comprehensive overview of the progress made in understanding cell death mechanisms during development using the Drosophila model system. The manuscript is well-written and easy to follow. However, the following concerns should be addressed before publication in Cells.

1)    The authors should provide a more thorough discussion on how these findings in Drosophila could potentially have implications for human health and disease. Although the fly model system is widely utilized for elucidating fundamental biological processes, it is important to emphasize the potential transferability of these findings to human biology, particularly in the context of diseases like cancer, where aberrations in cell death mechanisms play a pivotal role.

2)    The review could benefit from a more detailed discussion on the tools and techniques commonly used in Drosophila cell death-related research. This would be particularly helpful for non-fly researchers who are new to the field and will have a broader general interest.
